# Exceptional origin activation revealed by comparative analysis in two laboratory yeast strains

**Ishita Joshi[1]☯, Jie Peng[1]☯, Gina Alvino[2], Elizabeth Kwan[2], Wenyi Feng[1]***

**1** Department of Biochemistry and Molecular Biology, SUNY Upstate Medical University, Syracuse, New York, United States of America, **2** Department of Genome Sciences, University of Washington, Seattle, Washington, United States of America

☯ These authors contributed equally to this work.
* fengw@upstate.edu

**Data Availability Statement:** All relevant data are within the paper and its Supporting information files. Raw microarray data and DNA sequences of the A364a genome have been deposited in Gene Omnibus Database under the accession number

## Abstract

We performed a comparative analysis of replication origin activation by genome-wide single-stranded DNA mapping in two yeast strains challenged by hydroxyurea, an inhibitor of the ribonucleotide reductase. We gained understanding of the impact on origin activation by three factors: S-phase checkpoint control, DNA sequence polymorphisms, and relative positioning of origin and transcription unit. Wild type W303 showed a significant reduction of fork progression accompanied by an elevated level of Rad53 phosphorylation as well as physical presence at origins compared to A364a. Moreover, a rad53K227A mutant in W303 activated more origins, accompanied by global reduction of ssDNA across all origins, compared to A364a. Sequence polymorphism in the consensus motifs of origins plays a minor role in determining strain-specific activity. Finally, we identified a new class of origins only active in checkpoint-proficient cells, which we named "Rad53-dependent origins". Our study presents a comprehensive list of differentially used origins and provide new insights into the mechanisms of origin activation.

## Introduction

Eukaryotic genomes are reliant on multiple initiation sites, called origins of replication, on each chromosome for complete duplication of the DNA in every cell cycle. Much of our knowledge concerning the location and usage of replication origins comes from studies using the budding yeast, *Saccharomyces cerevisiae*, where origins were initially identified in a genetic assay as autonomously replicating sequences (ARS) capable of supporting plasmid replication [1]. Within each ARS, an 11-basepair (bp) ARS-consensus sequence (ACS), WTTTATRTTTW, has been found to be essential for origin function [2–4]. The ACS serves as the binding site for a hexameric origin recognition complex (ORC) [5, 6]. The ORC-bound sites are then joined by Cdc6 and Cdt1, which further recruit the hexameric MCM2-7 DNA helicase to form a pre-replicative complex (pre-RC) [7, 8]. The pre-RC is then activated by additional protein factors to establish the bi-directional replication forks [9]. In the *S. cerevisiae* genome more than

GSE166733 and are immediately accessible to the public.

**Funding:** This study was funded by the National Institutes of Health grant 5R01 GM118799-01A1 awarded to W.F. The funder had no role in study design, data collection and analysis, decision to publish, or preparation of the manuscript.

**Competing interests:** The authors have declared that no competing interests exist.

12,000 ACSs have been found; however, only approximately 400 origins are used during a single cell cycle, suggesting cis-acting sequences other than the ACS function in specifying or influencing origin usage [10, 11]. Thus, *a priori* sequence polymorphism among different laboratory strains is expected to exert a direct impact on origin usage. Indeed, genetic variation in diverse laboratory strains can manifest in distinct physiological properties including the ability to adapt to growth in nutrient-deprived conditions, mating and sporulation efficiencies, filamentous growth, and transformation efficiency, etc. [12–15]. Simplistically, origin usage difference would be binary in nature, *i.e.*, an origin is capable of firing in one genetic background but not in another due to polymorphisms in essential regions of the origin, regardless of growth conditions. For instance, a SNP in the A364a strain created an origin, *ARS314.5*, that is absent from the YPH45 strain where ARSs on chromosome III were exhaustively characterized [16, 17]. Additionally, sequence polymorphism could alter the property of an origin. For instance, a 19-bp sequence in the *URA3* locus has been found to advance the time of origin activation (see below) through an unknown mechanism [18–20]. Finally, differences in origin usage could also manifest as differential response to changes in growth conditions and/or genetic mutations in origin regulation [21].

Studies using a replication timing assay based on the Meselson-Stahl density transfer method revealed a hierarchy of origin activation, which follows a temporal order in S phase [22–24]. This hierarchy is more starkly manifested in cells undergoing S phase in the presence of the ribonucleotide reductase inhibitor, hydroxyurea (HU), where initiation is only observed at origins that normally fire in roughly the first half of an unchallenged S phase but not at origins that would normally fire later in the unchallenged S phase [16, 25–27]. This hierarchy of origin activation is disrupted in cells carrying mutations in DNA replication checkpoint genes, such as *mec1* and *rad53*, in conjunction with replication stress by HU or other DNA damage-inducing drugs [16, 25–30]. Specifically, those origins that tend to be activated later in S phase in checkpoint-proficient cells are prematurely activated in the checkpoint mutants [16, 25, 31]. Mechanistically it has been shown that Rad53 inhibits late origin activation through phosphorylation of Dbf4 and Sld3 [32–34]. Consequently, virtually all origins fire in HU in checkpoint mutants, in contrast to wild type cells where only a subset of origins fire. This phenomenon permitted the classification of origins into two groups, "Rad53-unchecked" and "Rad53-checked" origins, based on their activation status (positive and negative, respectively) in a kinase-dead *rad53K227A* mutant versus a *RAD53* background, in the presence of HU [16]. While this mechanism of checkpoint control of origin activation appears to generally hold true among different strains, whether a given origin is subject to checkpoint control equally across diverse genetic background has not been assessed in a systematic manner.

In *S. cerevisiae*, origins predominantly reside in intergenic regions [35]. This feature is presumably the evolutionary outcome of minimizing replication-transcription conflict, which has precedents in both prokaryotic and eukaryotic genomes [36]. However, a study that systematically mapped Mcm2-7 binding sites in both mitotic and meiotic cells showed that 106 of 393 origins that contained Mcm2-7 binding sites were intragenic, *i.e.*, overlapping with the ORF or the promoter of a gene [37]. Twenty of these intragenic origins (19%) showed meiosis- or mitosis-specific Mcm2-7 loading. This differential origin activity is primarily due to gene expression during meiosis precluding origin activation and resulting in mitosis-specificity, and vice versa, underscoring transcription-replication incompatibility [37]. Nevertheless, 86 intragenic origins (81%) were apparently active regardless of meiotic or mitotic growth. How these intragenic origin activities are coordinated with transcription so as not to be obliterated by transcription is a fascinating problem. On the other hand, a recent study has shown that the ACS element(s) within the replication origin can pause or terminate RNA Pol II-mediated pervasive transcription, suggesting that intragenic origins might assume a more proactive role

than previously thought with regards to coordination with transcription [38]. The genetic determinants for the regulation of intragenic origins are not understood.

As outlined thus far, all the questions above would be effectively addressed by systematically comparing origin dynamics in different laboratory yeast strains where sequence conservation as well as variation have been well characterized. Yet, there is a conspicuous dearth of such studies. Therefore, we set out to systematically map and characterize origin usage in two commonly used strains, A364a and W303, using the genome-wide origin mapping method based on labeling of replication fork-associated single-stranded DNA (ssDNA) [16, 39]. Both A364a and W303 are closely related to S288C, the first strain to be whole-genome sequenced. A364a has been widely used for cell cycle studies [40]. There is also a wealth of information on replication dynamics both at chromosomal level and on a genome-wide scale in A364a [16, 19, 22, 24, 41]. W303 is a closer derivative of S288C than A364a [13] and is widely used for studies including DNA repair and ageing research [42, 43]. It has also been characterized for replication initiation factors (ORC and MCM) binding sites as well as replication dynamics using a variety of approaches [25, 44–46]. However, there has not been a study that systematically analyzes origin activities using the same technological platform in these two well characterized strains or--indeed--in any two strains. Therefore, we set out to fill this gap with the current study. Because the presence of HU restricts the pool of dNTPs and thereby limits replication fork progression, the genomic locations of S-phase specific ssDNA accumulation in HU mark the locations of active origins. We introduced a kinase-dead and checkpoint-deficient K227A mutation in the *RAD53* gene in each of the two strains and compared origin usage in wild type and the *rad53* mutant cells. We also integrated whole genome sequencing data to identify single nucleotide polymorphisms (SNPs) in A364a and W303. Our study demonstrates that by and large the replication checkpoint exerts control over origin activation similarly in the two strains. However, W303 exhibited different replication dynamics than A364a in that 1) replication forks were more constrained in W303 cells containing a wild-type *RAD53*; and 2) more origins were activated in W303 upon checkpoint inactivation due to the *rad53K227A* mutation. Moreover, we did find strain-specific origin usage and the potential contributing SNPs, thus enabling future studies to identify the regulatory mechanisms for origin usage. Finally, we identified a new class of origins that are only active in cells with an intact checkpoint. These origins tend to reside in regions overlapping open reading frames (ORFs). However, the lack of origin activation in the *rad53* mutant cannot be explained by increased transcription. Therefore, we suggest that origin activation at these specific locations requires an intact Rad53 kinase at an undefined step.

## Results

### More origins of replication were activated in the W303 background than in A364a

We surveyed origin activity in cells containing wild type *RAD53* or a *rad53K227A* kinase-dead mutation in both genetic backgrounds, A364a and W303, using a previously described method of mapping ssDNA gaps at the replication fork stalled by HU [16, 39]. Briefly, yeast cells were embedded in agarose plugs and then spheroplasted to reveal the chromosomal DNA while protecting the single-stranded gaps therein. The ssDNA would then serve as the template for random-primed DNA synthesis without denaturation to separate the duplex DNA. The DNA from an S phase sample and a G1 control sample were differentially labeled with Cy3- and Cy5-dNTP, respectively, and then co-hybridized to a microarray to reveal the locations of S-phase specific ssDNA (Fig 1A). Two biological replicates were performed for each sample. In all cases the biological replicates exhibited high concordance (R> = 0.97), underscoring the

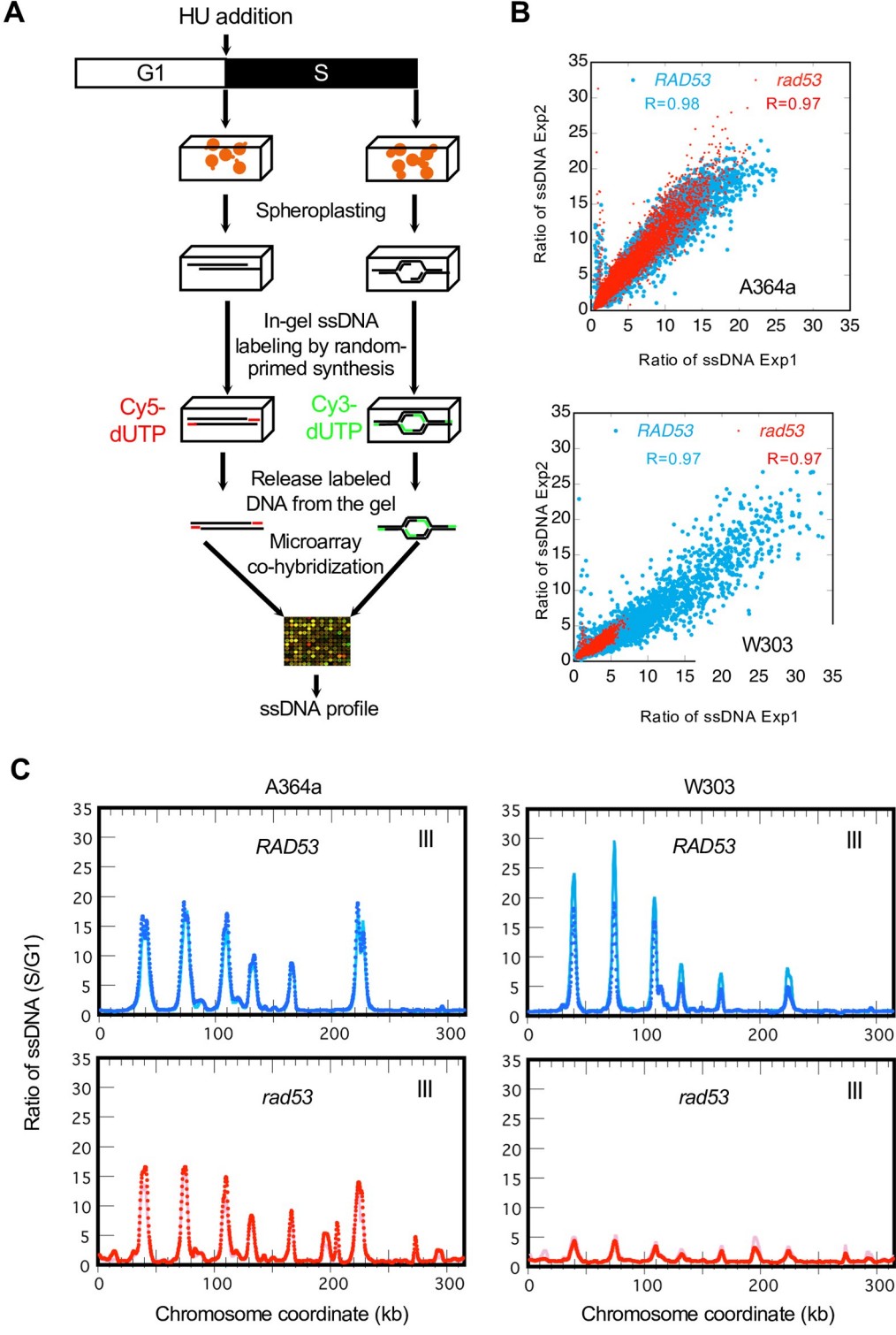

**Fig 1. Genome-wide ssDNA mapping in two laboratory yeast strains. (A)** Schematic experimental procedures of ssDNA mapping by microarray. **(B)** Reproducibility of ssDNA ratios (S/G1) from two independent experiments. Pearson correlation co-efficient values are shown. **(C)** Normalized ssDNA profiles of ChrIII (ssDNA ratios of S/G1 plotted against chromosome coordinates in kb) for *RAD53* and *rad53-K227A* cells in the A364a (left) and the W303 (right) background from one of the two replicate experiments (Exp 1).

reproducibility of the ssDNA measurements (Fig 1B). Chromosomal plots of ssDNA profiles were nearly identical between replicates, revealing ssDNA peaks at discrete locations of the chromosomes (Fig 1C). See all chromosome plots for Exp 1 in S1 Fig in S2 File.

Next, we developed a pipeline to systematically query origin firing status based on ssDNA levels at each of 626 "confirmed" and "likely" origins queried at OriDB (http://cerevisiae.oridb. org/). The two basic components of the pipeline were 1) ssDNA peak calling by local maxima identification and 2) assigning the ssDNA signals to a particular origin. A typical origin in wild type yeast sends off a pair of replication forks in opposite directions, resulting in a ssDNA profile of two adjacent peaks flanking the center of an origin. However, the same ssDNA profile can be mistaken for two adjacent origins each producing a single ssDNA peak. Therefore, a major challenge was to accurately differentiate these two possibilities and assign the ssDNA signals to the correct origin(s). To do so we employed three rigorously chosen and stringent parameters (Materials and methods). First, only those origins with a ssDNA peak amplitude greater than three standard deviations above background in both replicate experiments were considered. Second, we uniformly defined origin size to be 4 kb to enable optimal ssDNA peak-origin association in an unbiased fashion. Third, we determined a minimal distance of 1.75 kb between two adjacent ssDNA peaks to be accepted as two separate origins. As detailed in Materials and Methods, our algorithm would incur an estimated 4% false negative rate due to the chosen 1.75 kb minimal inter-peak distance. Additionally, origins with moderate level of ssDNA might also elude detection, as in the case of *ARS813* (see below). Overall, we identified 362 and 405 consensus origins (detected in both replicate experiments) for A364a and W303, respectively (Fig 2A and S1 File). Altogether there were 419 unique origins of which the majority (348) were shared by A364a and W303 and the rest were activated in a strain-specific manner. Each of the 419 origins was then classified into one of three categories based on the comparison between *RAD53* and the *rad53K227A* mutant (see below).

## The extent of ssDNA formation across the origin is lower in the W303 background than in A364a

We first compared the extent of ssDNA progression over all 419 origins in each strain by visualizing them on a heat map. The results showed that the extent of ssDNA formation in both *rad53* mutants in the A364a and W303 background was reduced compared to their *RAD53* counterparts (Fig 2B). This observation was consistent with *rad53* cells accumulating aberrant replication fork intermediates when encountering HU, ultimately resulting in replication fork collapse or the failure to resume synthesis [26, 47]. Additionally, both the *RAD53* and mutant versions of the W303 background showed reduction of ssDNA compared to their A364a counterparts (Fig 2B). Similar conclusions were drawn from generating aggregated ssDNA profiles across a 20 kb window centering on all 419 origins in each experiment (S1A Fig in S2 File) followed by calculating area under the curve (AUC) of the ssDNA peaks and averaging two replicate experiments for each sample (Fig 2C). Additionally, *rad53* mutant exhibited global reduction of ssDNA at origins compared to wild-type cells specifically in the W303, and not the A364a background (Fig 2C). We also calculated the average fork distance based on the width of the ssDNA peaks for each sample (S1B Fig in S2 File). Fork distance was significantly reduced in W303-*RAD53* cells than in A364a-*RAD53* cells (S1A & S1B Fig in S2 File). These results were not due to a difference in synchrony of the cell cultures as cell cycle analysis using budding index (S2A Fig in S2 File) or flow cytometry (S2B Fig in S2 File) both confirmed synchronous entry into S phase. We then asked if the reduced fork progression in W303 is attributable to a more active S-phase checkpoint. The results demonstrated that phospho-Rad53 (Rad53-P) was readily detected in WT cells in both A364a and W303 (Fig 2D). The checkpoint

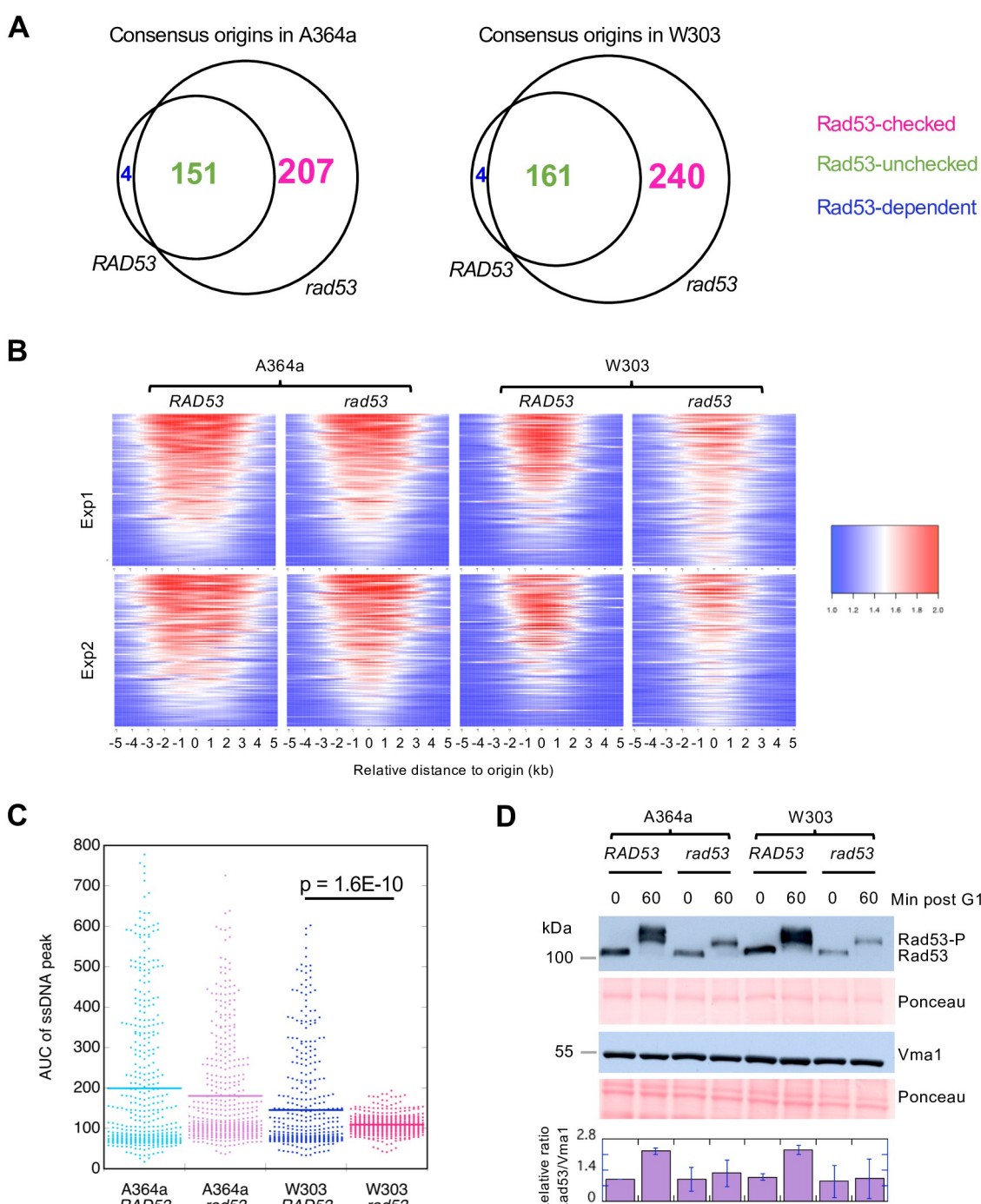

**Fig 2. Comparative analysis of origins of replication in the A364a and W303 background. (A)** Consensus origins identified by both biological replicate experiments in the A364a (left) and W303 (right) background, respectively, were divided into three classes of origins, "Rad53-unchecked", "Rad53-checked", and "Rad53-dependent", based on the comparison between *RAD53* and *rad53* cells. See text for detail. **(B)** Heatmaps of the normalized ssDNA ratios (compressed to a scale of 1 to 2) from the 419 unique origins from (A) in each of the four strains. **(C)** Quantification of average AUC values for the uncompressed ssDNA signals at the 419 origins in each strain from two experiments. **(D)** Western blots of Rad53 levels in cells at 0 or 60 min post release from G1 arrest into S phase in the presence of 200 mM HU, respectively. A cellular protein Vma1 was used as loading control. Overall protein levels were also controlled by Ponceau staining. The Rad53 level was quantified as the ratio of Rad53/Vma1. Each ratio was then normalized to the ratio from A364a_*RAD53* cells, which was set to 1. Data from three independent measurements were averaged and plotted. Error bars stand for standard deviation.

mutation reduced the levels of both Rad53K227A and Rad53K227A-P to a similar degree in both strain background (Fig 2D). These observations suggested that the checkpoint operated at similar levels in A364a and W303, and thus unlikely caused the global reduction of origin activation in the W303_*rad53* cells. We further asked if the potentially more active checkpoint caused a reduction of replication initiation factors at the origins in the W303 background. To do so, we introduced a 3X-HA epitope tag into the carboxyl-terminus of the endogenous *CDC45* locus. We then analyzed chromatin immunoprecipitation (ChIP) using the anti-HA antibody followed by quantitative PCR (qPCR) as a proxy to measure Cdc45 binding to select regions in the genome during synchronous entry into S phase in HU from G1 arrest. We first analyzed three regions on ChrIII: *ARS305* and *ARS306*, two early-firing origins; and *ARS313*, a relatively late-firing origin (Fig 3A). To control for nonspecific protein binding to the chromatin we also selected *ARS306*-dist, a non-origin control region 20 kb upstream of *ARS306* and 15 kb downstream of *ARS305*, where we expected little to no synthesis to occur during a synchronized S phase in HU. We then normalized the Cdc45 binding at each locus to that at this control locus, expressed as "Relative Enrichment" (Fig 3B). Upon initiation of replication Cdc45 was readily detected at *ARS305* and *ARS306* in WT cells (peaking at 20 min post release), consistent with unchecked origin firing in the presence of HU (Fig 3B). In the *rad53* mutants Cdc45 binding was significantly reduced at *ARS305* but not *ARS306*, suggesting that firing efficiency at *ARS305* was decreased. This phenotype at *ARS305* was later confirmed by two-dimensional gel electrophoresis (see below). At the late-firing *ARS313* there was relatively low level of Cdc45 binding in all strains, with the W303_*rad53* cells trending the highest but peaking later than at *ARS305* and *ARS306* (Fig 3B). The relatively higher level of Cdc45 binding at *ARS313* in W303_*rad53* cells was unexpected and suggested alternative mechanism for the observed global depression of origin activation in these cells. Therefore, we analyzed two additional late-firing origins, *ARS603* and *ARS501*, and both origins showed similar Cdc45-binding patterns as *ARS313* (Fig 3A & 3B). We then asked if Rad53 itself has a higher level of presence at origins, thereby inhibiting origin firing, in the W303_*rad53* cells. To do so we performed Rad53-ChIP similarly as described above for Cdc45-ChIP except we focused on the 0 and 60-min time points post release from G1. The results showed that at 0 time point, W303_*rad53* cells exhibited higher level of Rad53 binding at all loci (though not statistically significant), and to a greater extent for the late-firing origins, than A364a_*rad53* cells (Fig 3C). These results suggested that increased level of Rad53, despite the elevated level of Cdc45, binding the origins in the W303_*rad53* strain might be the underlying cause for global repression of origin activation in these cells.

## Origins with different checkpoint status in A364a and W303

We next defined the unique status for each of the 419 origins based on its behavior in the four samples, *i.e.*, in *RAD53* and *rad53* cells of either A364a or W303 background. This resulted in the classification of the 419 origins into 15 categories (S1 File). First, we focused on the "Rad53-checked origins" and "Rad53-unchecked origins" (Fig 4A), previously defined as those origins only activated in *rad53* cells, hence subject to Rad53 checkpoint control, and those shared by *RAD53* and *rad53* cells, hence oblivious to Rad53 checkpoint control, respectively [16]. One hundred and forty-seven of 151 Rad53-unchecked origins in A364a (97%) were also identified as Rad53-unchecked origins in W303 (category 1 in Fig 4A). Similarly, 91% (185 of 207) of Rad53-checked origins in A364a were also found as Rad53-checked origin in W303 (category 2 in Fig 4A). Therefore, the vast majority of origins share the same checkpoint status despite differences in genetic background. Only twelve origins showed opposite classifications (sum of categories 3 and 4 in Fig 4A), which we referred to as "differential checkpoint status" origins.

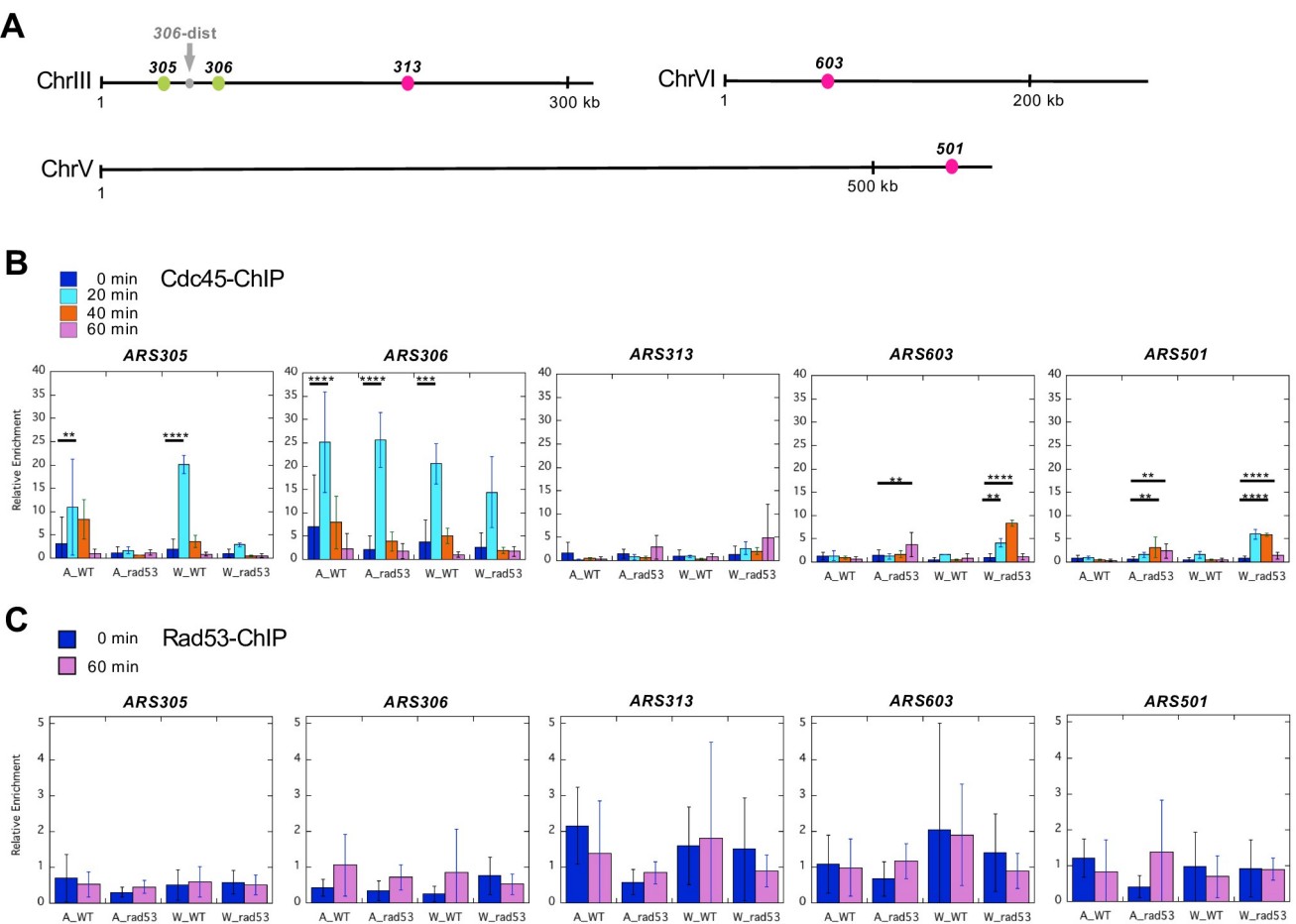

**Fig 3. ChIP-qPCR analysis of initiation factor binding to origins. (A)** Schematic diagram (drawn to scale) of the chromosomal loci used for qPCR analysis. The prefix "ARS" was dropped for simplicity. Note the *ARS306-dist* locus that is located 20 kb upstream of *ARS306* and 15 kb downstream of *ARS305* was used as a negative control region to normalize signals from all other loci. **(B)** Cdc45-ChIP was performed in cells collected at 0, 20, 40, and 60 min post release of G1-arrested cells into S phase in the presence of 200 mM HU. "A", A304a; "W", W303. Relative enrichment of the ChIP DNA signals was calculated as the ratio of %Input at a given locus over %Input at the control locus. %Input was calculated as the percentage of ChIP-DNA in input DNA, averaged from at least three replicate experiments. The error bars represent standard deviations. Statistical analysis was performed by one-way ANOVA followed by Tukey's multiple test. *, p<0.05; **, p<0.01; ***, p<0.001; ****, p<0.0001. **(C)** Relative enrichment of Rad53-ChIP DNA at the same loci as in (B). The analysis was done identically as in (B) except only cells collected at 0, and 60 min post G1 arrest/release were analyzed.

We verified these results by 2-D gel analysis, using the same conditions employed for the ssDNA mapping experiments, *i.e.*, harvesting cells synchronously released from G1 arrest into S phase in the presence of HU for 60 min. We did not use asynchronous cell populations to maximize the chance of detecting strain-specific differences as a result of HU-induced replication stall. We first validated *ARS305*, a Rad53-unchecked origin in both W303 and A364a. As expected, bubble arcs were detected in all samples, with or without an intact checkpoint (Fig 4B). Notably, the bubble arc signal in *rad53* cells in W303 was significantly reduced compared to other samples, confirming the observed global reduction of origin activation based on ssDNA mapping. We also analyzed *ARS813*, which was classified as a Rad53-checked origin in both A364a and W303 and yet showed a clear ssDNA signal in *RAD53* cells in A364a (Fig 4C). Indeed, bubble arcs were detected in this the A364a_*RAD53* cells (Fig 4C). Thus, *ARS813* was mis-classified as category 2 by the algorithm due to the moderate level of ssDNA (less than three standard deviations above background), and should fall in category 3. This brought the

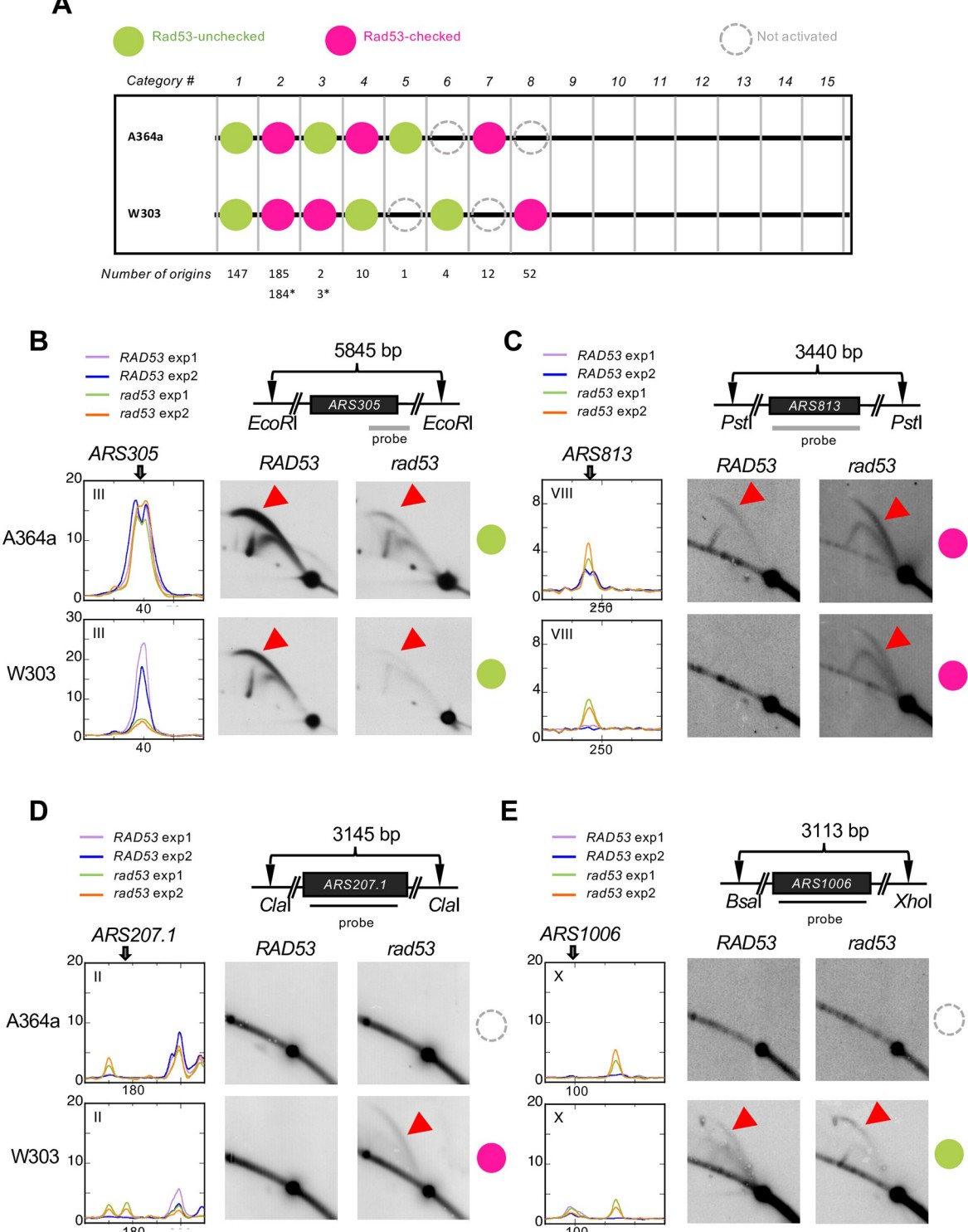

**Fig 4. Validation of Rad53-checked/unchecked origin firing status using 2-D agarose gel electrophoresis.** (**A**) Comparison of origin usage in the two strains leads to further classification of active origins into 15 categories. The number of origins in each category is listed below the diagram. The numbers with asterisks are the adjusted number after manual inspection. See text for detail. (**B-E**) DNA fragments containing select origins for analysis (*ARS305*, *ARS1006*, *ARS207.1*, and *ARS813*) were produced by restriction digestion with the indicated restriction enzymes. DNA probes are shown as black bars. Origin position is as indicated on the ssDNA profiles from two biological replicate experiments. "Bubble arcs" indicative of origin activation are indicated as red arrowheads. The origin classes were indicated using the key described in (**A**).

number of "differential checkpoint status" origins to 13 (sum of categories 3, with asterisk, and 4 in Fig 4A).

## Strain-specific usage of origins

We identified 71 (14 in A364a and 57 in W303) origins that were only activated in one of the two strains (sum of categories 5–8 in Fig 4A and categories 12 and 15 in Fig 6A, discussed later). We termed them "strain-specific origins" and also confirmed them by 2-D gel analysis. For instance, a bubble arc was detected at *ARS207.1* only in *rad53* cells in the W303 background, making it a W303-specific and Rad53-checked origin (Fig 4D). Similarly, a bubble arc was detected at *ARS1006* in both Rad53 and *rad53* cells in W303, but not in A364a, making *ARS1006* a W303-specific and Rad53-unchecked origin (Fig 4E).

We next asked if the strain-specific usage could be explained by sequence polymorphisms within these origins. We made use of the previously published contig sequences from the *de novo* assembly of whole genome sequencing data in W303 [15]. To identify SNPs in the W303 strain we performed BLAST searches using DNA sequences containing 71 strain-specific origins retrieved from the S288c reference genome against the contigs from W303. We also generated contigs from *de novo* assembly of whole genome sequencing data of an A364a strain and performed BLAST searches similarly. We found 32 origins with full-length sequence contigs in both genetic background based on OriDB coordinates. Eight of these 32 origins contain strain-specific polymorphisms, with five (*ARS201.5*, *ARS(5:109)*, *ARS810*, *ARS1233*, and *ARS (15:1023)*) in the A364a background and three (*ARS207.1*, *ARS221.5*, and *ARS1429*) in the W303 background (Fig 5A & 5B). All but one origin, *ARS1429*, were W303-specific and showed no activity in A364a. Next, we searched for ACS elements in these origins and asked to what extent the strain-specific origin usage was due to polymorphisms in the ACSs. Unexpectedly, none of the eight origins contained any perfect ACS (WTTTATRTTTW), despite multiple near matches, in the OriDB-curated ARS coordinates. *ARS221.5*, *ARS(5:109)* and *ARS1429* each contained a single degenerate ACS (WTTTA**Y**RTTTW) based on mutagenesis studies on *ARS307* [3], while the remaining five origins were devoid of even the degenerate ACS. We also checked for the presence of an extended ACS (EACS) previously proposed based on studies on *ARS309*, WWWWTTTAYRTTTWGTT [48], and only *ARS221.5* contained one such EACS. We then surveyed the ARS sequences from a high-resolution ARSseq study [49] curated at the Database of Eukaryotic ORIs (http://tubic.tju.edu.cn/deori/) to obtain more precise locations for these eight origins and looked for ACSs [50]. Unfortunately, only two origins (*ARS201.5* and *ARS1233*) were mapped by ARSseq. The average origin size based on ARSseq is significantly smaller than that based on OriDB (568 bp vs. 1274 bp), making it unlikely that we missed the consensus ACSs by using the OriDB coordinates. Indeed, the ARSseq core sequences for *ARS1233* were well within the boundaries defined by OriDB. *ARS201.5* ARSseq core sequences (28719–29138) extended beyond the region defined by OriDB (28941–29160) at the 5'-end. We searched the extended region and still found no consensus ACS. Therefore, it appeared that the strain-specific origins are devoid of consensus ACSs and thus are dependent on the near-matches of ACS elements to initiate replication. We then asked if the near-match ACSs were polymorphic between A364a and W303. Only *ARS201.5*, a W303-specific origin, contained a polymorphism (29064:T>A) within such a near-match ACS (AAAATAT CAAG, ACS is on the—strand) in the A364a background (Fig 5A). Note the 29064:T>A change actually improved the ACS element in A364a. None of the other polymorphisms occurred in a near-match ACS, nor did they create a new ACS. Taken together these data suggest that strain-specific origin usage cannot be accounted for by SNPs in the ACS elements.

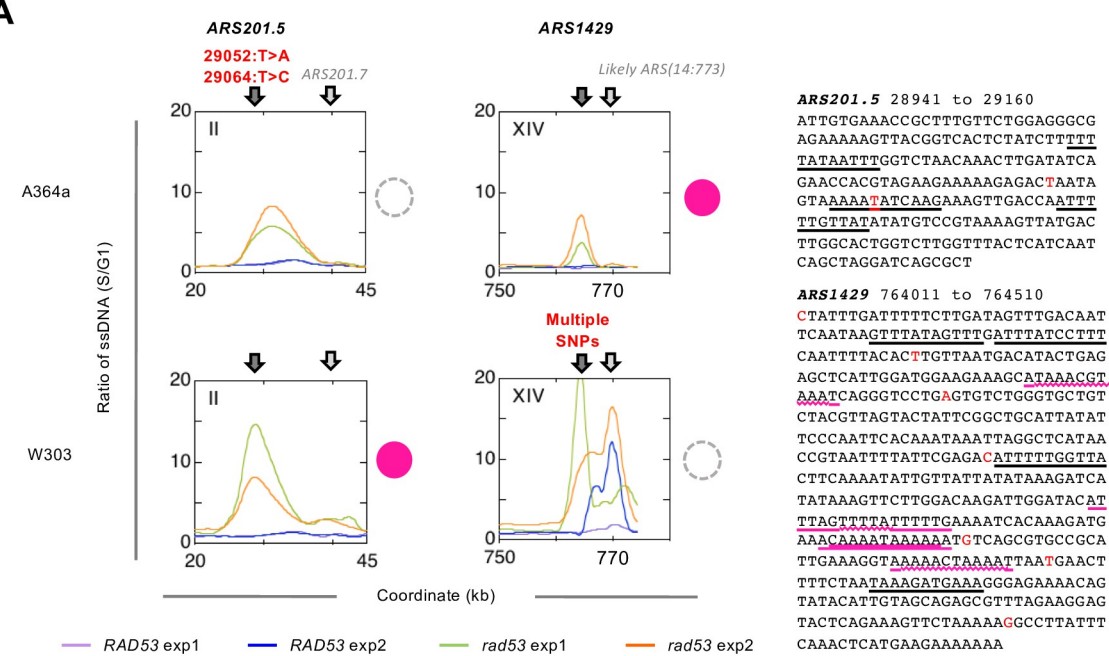

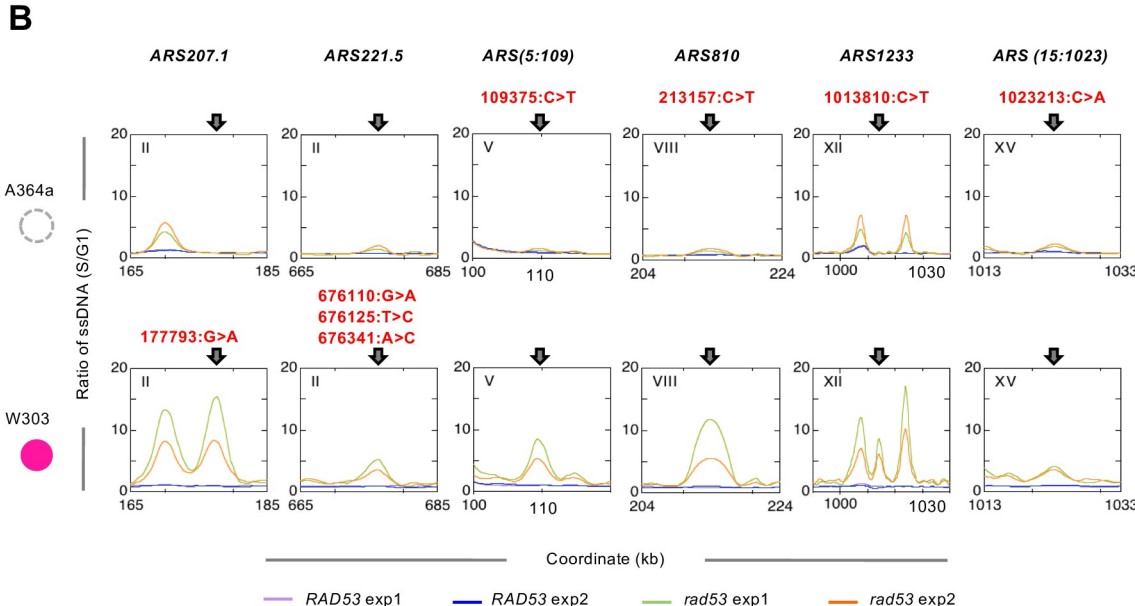

**Fig 5. Examples of strain-specific origins with polymorphisms.** The corresponding origin status is color coded same as in Fig 4. **(A)** Two origins with polymorphisms in A364a or W303 background resulting in a shift of initiation sites from the canonical origins. The SNPs (red font) are indicated in the ARS regions based on OriDB curation on the right. The near-matches of ACS elements are underlined in black or, in the case of overlapping elements, in magenta. The sequences shared by overlapping elements are underlined by wavy magenta lines. **(B)** Six origins that show binary pattern of activation in A364a but not W303 or *vice versa*, without shifting initiation sites.

Additionally, closer scrutiny of the ssDNA profiles revealed two distinct patterns of strain-specific usage of origins. In one of them a more binary origin activity was observed, *i.e.*, a ssDNA peak at the specific origin was present in W303 and absent in A364a (Fig 5B). In the other pattern, two origins, *ARS201.7* and *ARS1429*, which were inactive in one strain, appeared to have shifted the activation site downstream from the origin in the other strain (Fig 5A). In both cases multiple SNPs were found in the genetic background where the origin was inactive (Fig 5A). For instance, *ARS1429* was deemed inactive in W303 by the algorithm—a well-defined ssDNA peak at *ARS1429* was only visible in one of the two experiments in W303—and consequently initiation took place from a likely origin (*ARS(14:773)*) in that experiment (Fig 5A, bottom right plot). This likely origin was not activated in A364a (Fig 5A, top right plot). Similarly, the ssDNA peak appeared to have shifted downstream from *ARS201.5* to a location between *ARS201.5* and *ARS201.7* in the A364a background (Fig 5A, top left plot), while the ssDNA peak was centered on *ARS201.5* in the W303 background (Fig 5A, bottom left plot). No ARS, including dubious ARSs in the OriDB, was found in this region. Therefore, it appeared that the polymorphisms at these two origins in the corresponding genetic background rendered initiation to take place elsewhere.

## Rad53-dependent origins

Finally, we discovered six origins that were only active in *RAD53* and not the checkpoint mutant cells in at least of one of the two strain background (Fig 6A). We termed this novel class of origins "Rad53-dependent origins". Two of these origins showed strain-specific usage, *ARS1016* (W303) and *ARS1618.5* (A364a), as discussed above (categories 12 and 15 in Fig 6A). Another two origins, *ARS1206.5* and *ARS207.8*, showed Rad53-dependence in only one background (categories 11 and 13 in Fig 6A). Finally, two origins were consistently identified as Rad53-dependent in both strain background: *ARS(13:269)* and *ARS(16:560)* (category 9 in Fig 6A). We first focused on the two origins that were consistently identified as Rad53-dependent in both A364a and W303 and validated them by 2-D gel. Indeed, both *ARS(13:269)* and *ARS(16:560)* appeared to be only active in *RAD53*, and not in *rad53* cells (Fig 6B & 6C). This is a new class of origins that have not been described in literature. The only discernable feature associated with these origins was their proximity to the nearest ORF. Five of these six Rad53-dependent origins, with the exception of *ARS1206.5*, overlap with an ORF. This led us to ask if the lack of origin activity in the *rad53* cells was due to higher level of transcription at these loci in *rad53* cells than in the *RAD53* cells. We analyzed the ratio of gene expression in S phase over G1 control (S/G1) at multiple regions spanning the two origins validated by 2D gel, *ARS(13:269)* and *ARS(16:560)* by real-time reverse transcription-PCR. Two regions downstream of *CDC5*, P1 and P2, the latter of which overlaps with *ARS(13:269)*, both showed higher level of transcription in *rad53* cells regardless of strain background (Fig 6D). Remarkably, P3 that overlaps both the origin and *CDC5* and is the closest to the initiation point of *ARS(13:269)*, showed higher level of transcription in *RAD53* than *rad53* cells (Fig 6D). Similarly, transcription was higher in *RAD53* cell than in *rad53* cells, though less evident for the W303 background, at the P4 locus which bookends with the origin but is enclosed in *CDC5* (Fig 6D). Across *ARS(16:560)*, all loci from P1 to P6 showed higher level of transcription in *RAD53* cells than in the *rad53* mutant (Fig 6D). These results suggested that the lack of origin activation in *rad53* cells is not due to high level of transcription. We also analyzed a positive control at *DUN1*, a checkpoint kinase whose expression is induced by HU during S phase [51, 52]. The results showed that *DUN1* expression was indeed higher in *RAD53* cells compared to the *rad53* mutant, in both strain background (S3 Fig in S2 File). In contrast, GADPH expression was similar between *RAD53* and

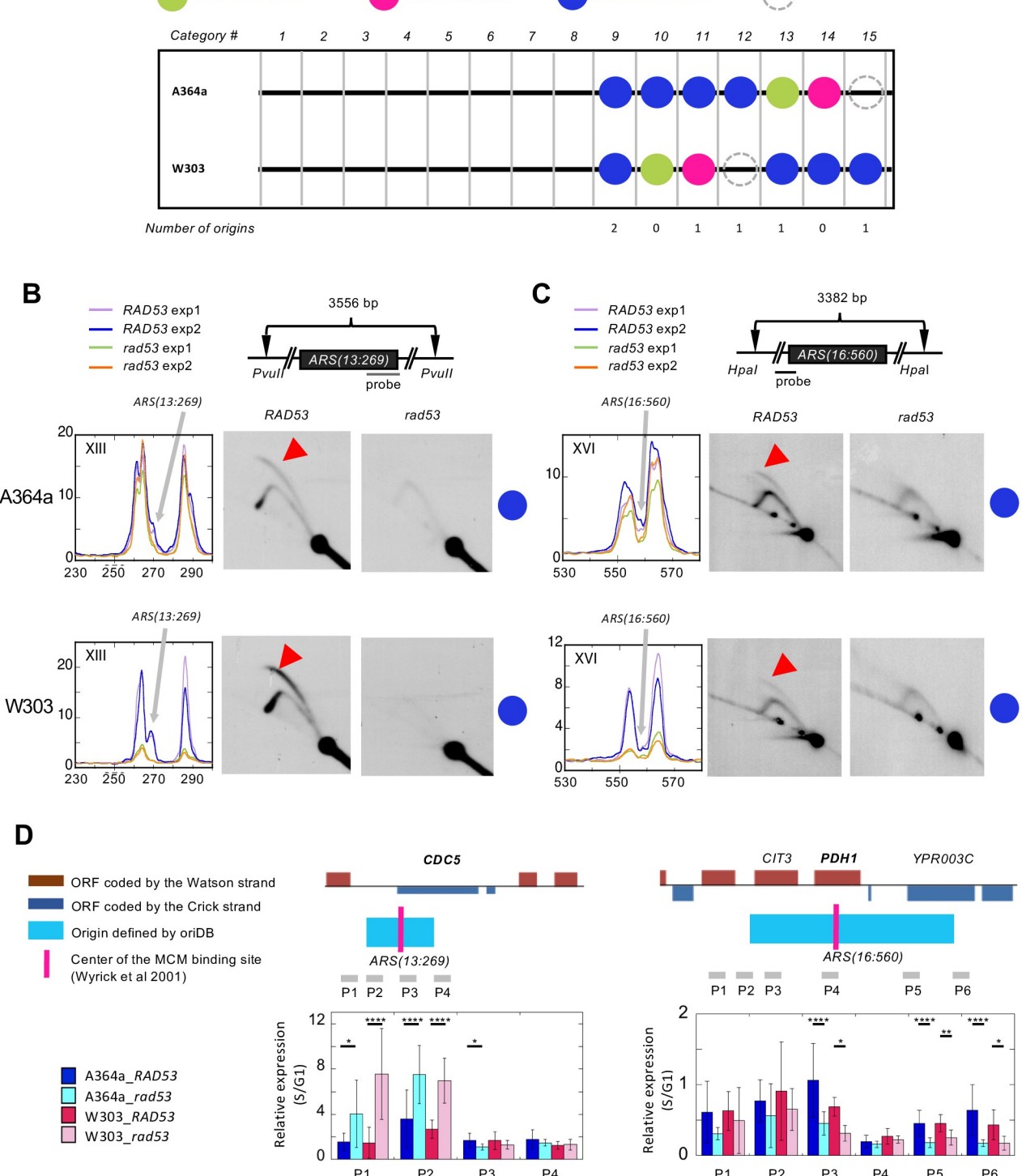

**Fig 6. Rad53-dependent origins. (A)** Numbers of Rad53-dependent origins parsed into the shown categories. **(B & C)** 2-D gel validation of *ARS (13:269)* and *ARS(16:560)* performed similarly as described in Fig 4. **(D)** Relative gene expression levels (S/G1) at genomic loci of Rad53-dependent origins that overlap with ORFs. The relative locations of the origins and location of ORC binding sites are derived from OriDB. The ORF that is nearest to the ORC binding site is bolded in each locus. The primers (P#) used for RT-qPCR are shown as a light grey bar, not drawn to scale.

*rad53* cells in the W303 background, with moderate increase in the *RAD53* cells in the
A364a background (S3 Fig in S2 File).

## Discussion

We conducted a comparative analysis of origin activation upon DNA replication stress by HU,
using genomic ssDNA mapping in two common laboratory yeast strains. By coupling with
whole genome sequence information our experimental design offered the opportunity to
directly test the impact of sequence polymorphism on origin usage as well as on the checkpoint
response to replication stress. The ssDNA mapping results were corroborated by 2-D gel anal-
ysis. Consistent with previous findings more origins in the genome are under the control of
the Rad53-mediated checkpoint in HU, *i.e.*, do not fire in HU in wild type *RAD53* cells, than
origins that are impervious to the checkpoint in both genetic backgrounds. The vast majority
of origins (331) share the same checkpoint status between the two genetic backgrounds. This
suggests that the replication checkpoint by and large exerts control over the same origins in
A364a vs. W303, provided the origin is competent for activation. Only 13 origins showed
opposite checkpoint status in the two genetic backgrounds. Below we discuss the key observa-
tions with regards to the differences in genetic background.

We detected 71 origins that showed strain-specific usage, 32 of which were fully sequenced
in both the W303 and A364a background. Comparative sequence analyses of 8 origins with
strain-specific polymorphisms indicated that none of the strain-specific origin usage could be
explained by polymorphisms in the ACS elements. In fact, none of these 8 origins contained a
perfect ACS, only near-matches of an ACS with 1–2 mismatches. This observation suggests
that a perfect ACS, or even one with a single base mismatch, is not required for activation of
these strain-specific origins. Instead, the presence of multiple near-match ACSs acting in con-
cert is probably sufficient for origin activation. This theory had been articulated previously by
Van Houten and Newlon, as they noted that several ARSs including the efficient *ARS606* and
*ARS305* only contain near matches of ACS that were in fact nonfunctional in the *ARS307* con-
text [3]. The vast majority of the SNPs, with only one exception, were located even outside
these near-matches ACSs. Finally, the remaining twenty-four strain-specific origins are devoid
of polymorphisms within the origin, suggesting that distal elements from the current known
origin regions might play a role in conferring strain-specific usage. Therefore, our results
defined a paradigm of sequence characteristics for origins in which a perfect ORC binding site
may be preferred but not required. This feature might explain why higher eukaryotes including
*Schizosaccharomyces pombe* and metazoans do not appear to have consensus sequence ele-
ments in their origins and instead rely on alternative mechanisms to concentrate ORCs upon
which to initiate origin activation.

Compared to the A364a background there were more activated origins in W303, specifically
in the Rad53-checked category. The W303 cells also showed a smaller distance covered by rep-
lication forks than A364a_*RAD53* cells. Arguably the most striking phenotype associated with
the W303 background is that W303_*rad53* cells showed global reduction of origin activation.
ChIP-qPCR analysis revealed that this phenotype was not due to decreased level of initiation
protein (Cdc45) binding, but rather probably due to increased Rad53 binding at origins. We
speculate that these results could be explained by the multiple roles of Rad53 in regulating ori-
gin activation and replication fork progression during HU exposure. In addition to down-reg-
ulating late origin activation, it has been shown that Rad53 is also important for coordinating
leading strand synthesis with DNA unwinding [53]. Moreover, a recent study reported that
Rad53 limits excessive unwinding of DNA at replication forks by the replicative helicase CMG
(Cdc45-MCM-GINS) complex [54]. Therefore, the net impact of Rad53 at origins during

initiation and subsequent fork movement is likely difficult to delineate. Nevertheless, our data provided a clue to how increased Rad53 binding at W303_*rad53* origins may underlie global suppression of origin activation. It would be of interest to determine if titrating the Rad53 level can modulate the number of activated origins within the same strain.

Finally, we discovered a new class of "Rad53-dependent Origins". The only recognizable feature of these origins is their proximity to an ORF. In the two validated Rad53-dependent origins in both A364a and W303, *ARS(13:269)* and *ARS(16:560)*, the MCM binding site is located inside *CDC5* and *PDH1*, respectively. Surprisingly, origin activity did not appear to correlate with low level of transcription, contrary to the idea that transcription interferes with origin activity. In fact, the gene expression level of *CDC5* was higher in *RAD53* cells than in *rad53* cells (Fig 6D). Thus, it appears that transcription does not preclude the activities of these Rad53-dependent origins. Intragenic origins in the yeast genome are relatively less common, with *ARS604* and *ARS605* being the primary examples [35]. In both cases it was thought that transcription abolishes origin activity, constitutively at *ARS604* (inactive in both mitosis and meiosis) and meiotically at *ARS605* (only active during mitosis). However, this notion may require further scrutiny. First, *ARS604* activity has been detected in checkpoint mutants previously [25] and in the current study (S1 File), calling into question the inactive status that has been associated with this origin. Moreover, MCM binding was detected at *ARS605* in meiosis [37] despite the conclusion that it is inactive during meiosis [55]. Therefore, we suggest that activation of intragenic origins is more common than previously thought, at least in the context of replication stress in mitotic cells. Nevertheless, we note that in almost every case of the "Rad53-dependent origins", there is an active origin close-by. It is possible that the RAD53-dependent origins have a higher probability to fire because the nearby origin is more likely to be inactivated by transcription. In *rad53* cells, where transcription is reduced, the adjacent origin becomes more active, rendering the Rad53-dependent origin passively replicated. However, the underlying assumption for this explanation would be that low level transcription selectively impacts the adjacent origin, rather than the Rad53-dependent origin itself. For these reasons we believe that a more probable explanation for the Rad53-dependent nature of these origins is the requirement of an intact Rad53 checkpoint function. Whether Rad53 is required for establishing or maintaining the pre-Replication Complex assembly at these genomic locales remains to be determined.

In summary, our study provides a comprehensive catalogue of active origins in two common lab strains, which will assist future phylogenetic characterizations of replication origins in yeasts as well as our general understanding of the mechanisms of origin usage in eukaryotic organisms.

## Materials and methods

### Yeast strains

Yeast strains used in this study were derived from A139 (*MATa RAD5 bar1::LEU2 can1-100 ade2-1::ADE2 his3-11,15 leu2-3,112 trp1-1::TRP1 ura3-1::URA3*) [56], a *RAD5* derivative of W303 [43] or HM14-3a (*MATa bar1 trp1-289 leu2-3,112 his6*) in the A364a background [57]. The *rad53K227A* mutation was introduced into the *RAD53* gene locus by gene replacement as described [58]. For ChIP studies, the *CDC45* and *RAD53* gene loci in each of the four strains were separately introduced with a triple (3X)-HA (hemagglutinin) and a fused 2X-HA-6X-HIS (histidine) epitope, respectively, at the carboxyl termini. The plasmids containing the tagged CDC45 (pRS405-CDC45-3X-HA) and RAD53 (pRS306-RAD53-2X-HA-6X-HIS) were gifted by the Bell Lab at MIT, the Brewer lab at University of Washington and the Bedalov lab at Fred Hutchinson Cancer Research Center, respectively. They were used to integrate the epitope-

tagged CDC45 and RAD53, into A364a strains, by transforming cells with plasmids linearized with *Nco*I and *Hpa*I, respectively. The *CDC45-3X-HA* construct was subcloned into pRS303 at the *Sac*I and *Apa*I restriction sites, yielding pRS303-CDC45-3X-HA. The *RAD53-2X-HA-6X-HIS* construct was subcloned into pRS303 at the *BamH*I and *Xho*I restriction sites, generating pRS303-RAD53-2X-HA-6X-HIS. These two plasmids were used to integrate the epitope-tagged CDC45 and RAD53, into W303 strains, by transforming cells with plasmids linearized with *Nco*I and *Hpa*I, respectively. Integrants were selected based on leucine (pRS405), uracil (pRS306) and histidine (pRS303) prototrophy, respectively. Expression of the epitope-tagged proteins were confirmed by western blot.

## Genomic ssDNA mapping by microarrays

Detailed procedures for ssDNA labeling, microarray hybridization and data analysis were described previously [39]. Briefly, cell cultures were grown to an $OD_{600}$ of 0.25~0.3 in YPD medium, followed by G1 arrest with 200 nM α-factor. Pronase (0.02 mg/mL) was used to synchronously release cells into S phase in the presence of 200 mM hydroxyurea (HU). G1 control and S phase samples were collected prior to cell cycle release and after 1 h treatment of HU, respectively. Three-hundred ml of cells from each sample were collected and spheroplasted in agarose plugs for ssDNA labeling. Differentially labeled G1 (Cy5-dUTP) and S phase (Cy3-dUTP) DNA were co-hybridized onto Agilent Yeast Whole Genome ChIP-on-chip 4 × 44K (G4493A) microarrays and the data were extracted by the Agilent Feature Extraction Software (v9.5.1). The relative quantity of ssDNA at a given genomic locus was calculated as the ratio of the fluorescent signal from the S phase sample to that of the G1 control, followed by Loess-smoothing over a 6-kb window at a step size of 250 bp.

## Flow cytometry

Cells were grown identically as described above. One-ml of cells were collected from logarithmically grown cell culture (asynchronous control sample), and at 0, 15, 30, 60, 90, and 120 min following synchronous release from α-factor arrest into S phase in the presence of 200 mM HU. Cell fixation and processing for flow cytometry were performed as previously described [56].

## Whole cell lysate preparation, SDS-PAGE and western blotting

Protein lysate preparation was adapted from the Zegerman lab protocol [59]. For each sample/condition, $10^8$ cells were collected. Cells were pelleted by centrifugation at 3200 rpm for 5 min before quick freezing on dry ice. To prepare protein lysates, frozen cell pellets were thawed in 200 μl of 20% TCA and transferred to a rubber seal screw capped 1.5-ml Eppendorf tube. To each thawed sample, 400 μl of glass beads were added before vigorous agitation in a bead beater homogenizer in the cold room, twice for 30 sec each with a 45 sec rest between. The beads were then resuspended in 400 μl of 5% TCA and the supernatant was transferred to a fresh Eppendorf tube. The beads were then washed with 400 μl of 5% TCA, and the wash was pooled with the first supernatant. The pooled supernatant was centrifuged for 2 min at 13200 rpm and the pellet was resuspended in 200 μl of 1x Laemmli buffer containing β-mercaptoethanol and 50 μl of 1M Tris base (to increase the pH). Protein lysates were boiled at 98°C on a heat block for 10 min, cooled on ice, and centrifuged at 13200 rpm for 2 min to clear the lysate. Forty μL of cell lysate was loaded on an 7.5% SDS-PAGE gel and electrophoresed at 100 V until the 100 kDa marker reached the dye-front for Rad53 gels and 35 kDa mark for Vma1 gels. Following transfer to 0.45-μm nitrocellulose membrane using standard procedure, the blots were blocked for 1 h at RT with 5% dry milk in 1x TBS-T, then hybridized overnight at

4°C first with 1:1000 Rabbit anti-Rad53 (Abcam #104232) or 1:4000 anti-Vma1 (gifted from the Kane lab), followed by 1:10000 anti-Rabbit (Invitrogen #31460) and 1:5000 anti-Mouse (Invitrogen #31430) for 1 h at RT, respectively. Blots were washed thrice with 5% TBS-T for 10 min each time. Post auto-radiography using ECL detection reagent (BioRad), the membranes were washed with $dH_2O$, then incubated with Ponceau Stain Solution (0.1% Ponceau, 5% Acetic Acid) for 5–10 min before photographing.

### Identification of significant ssDNA peaks

We first identified those ssDNA peaks with significant height in each sample by a Python module, PeakUtils 1.1.0 (https://pypi.python.org/pypi/PeakUtils). This module requires a user-input parameter, the minimal distance between two adjacent peaks. The size of this minimal distance is inversely correlated with the total number of ssDNA peaks identified, as fewer close-by peaks would be simultaneously called as peaks with increasing minimal distance. We determined the median inter-peak distance using a range of minimal distance set from 0.5 to 2.75 kb (S1 Table in S2 File) and compared to the inter-origin distance (~27 kb and ~15 kb for 410 confirmed origins and for 626 confirmed and likely origins, respectively). We then asked in each of the six groups how many ssDNA peaks were associated with the 626 origins, *i.e.*, the ssDNA peak summit was located within the origin region (S2 Table in S2 File, see below for detailed definition for origin regions). We chose a minimal distance of 1.75 kb between adjacent peaks as the most appropriate based on the following considerations. First, in this group the average inter-peak distance from two biological replicates of *rad53* cells (where we expect virtually all origins to fire) was 20.75 kb in both A364a and W303, which fell well within the expected range for inter-origin distance (15 to 27 kb as described above). Second, by increasing the minimal distance from 0.5 to 1.75 kb we significantly decreased ambiguous assignment of a ssDNA peak to multiple (>2) closely spaced origins while maintaining correct assignment of two "split peaks" to those known early/efficient origins with inferred bi-directional fork movement. For instance, using Experiment #1 for *rad53* cells in A364a as a training data set we found all 32 cases where a single origin was associated with 2 ssDNA peaks appeared to be due to bi-directional forks instead of close proximity to another origin (S4 Fig in S2 File). However, further increasing the minimal inter-peak distance to 2.25 kb would result in false negatives as in the case of *ARS807*, because only one of the two split peaks would be associated with *ARS807*. Using the 1.75 kb minimal distance means those origins that are less than 1.75 kb away from each other would be potentially false negatives. Among the 626 confirmed and likely origins, 14 pairs of origins involving 26 unique origins have <1.75 kb inter-origin distance. Therefore, we estimated the false negative rate to be at lease 4% (26/626).

Another key determinant for ssDNA peak-origin association was the size of an origin. To avoid introducing potential bias due to differential origin sizes reported in OriDB, we defined each of the 626 origins as a region with a uniform size centering on the mid-point chromosome coordinate given by OriDB and we incrementally tested a range between 1 and 6 kb (S2 Table in S2 File). We chose 4 kb as the optimal size for two reasons. First, previous study has shown that the "replication bubble" size, or the overall distance that a pair of bi-directional replication forks would have covered in *RAD53* cells treated with 200 mM HU for 1 h, was ~4.2 kb [26]. Second, setting origin size at 3 kb only identified 16 origins with bi-directional replication forks (compared to 32 at the 4 kb-origin size), based on the training data set of Experiment #1 for *rad53* cells in A364a (S2 Table in S2 File). In contrast, setting origin size at 5 kb enabled correct identification of 6 more origins (3 with bi-directional forks and 3 with single peaks), but also incurred 6 false positives (S2 Table in S2 File). Therefore, we chose the 4 kb origin size for further analysis.

The Lowess-smoothed ssDNA values from each experiment were input into the INDEXES function of the PeakUtils 1.1.0 module with a 1.75 kb minimal inter-peak distance. The amplitude index (0–1 in value) of a ssDNA peak was calculated as

$$(\text{ssDNA} - \text{Min})/(\text{Max} - \text{Min}),$$

where Min and Max are the minimal and the maximal ssDNA value in the genome, respectively. A ssDNA peak amplitude index must exceed a threshold determined as follows for it to be considered significant:

$$\text{Threshold} = (\text{Med} + 3\delta - \text{Min})/(\text{Max} - \text{Min}),$$

where Med is the median ssDNA value of the given sample, and $\delta$ is the standard deviation of all ssDNA values below Med.

## Origin identification by their association with significant ssDNA peaks

The mid-point locations of 626 confirmed and likely origins were obtained from OriDB (http://cerevisiae.oridb.org/index.php). All origins were then defined as a 4 kb region centering on the mid-point. A significant ssDNA peak was determined to be associated with an origin if its location had at least 1 bp overlap with the origin region, using the INTERSECT function in BEDtools [60]. An origin is considered active if it is associated with a significant ssDNA peak (defined in the previous section) in both biological replicate experiments.

## Two-dimensional (2-D) agarose gel electrophoresis

Three hundred ml of cells were grown to an $OD_{600}$ of 0.25 ~ 0.3 in YPD medium and then synchronized in G1 with 200 nM α-factor. HU was added to a final concentration of 200 mM and then pronase was added at 0.02 mg/ml to release cells from α-factor arrest into the S phase. Cells were collected after 1 h. Genomic DNA was prepared according to one of two methods. In the first, cells were lysed and DNA purified according to the 'NIB-n-grab' method (http://fangman-brewer.genetics.washington.edu/nib-n-grab.html) and processed for 2-D gel analysis as described [18]. In the second method, cells from 100 ml of culture was collected and embedded in agarose before spheroplasts were prepared as described for ssDNA mapping. Each agarose plug was then washed with 5 ml of TE 0.1 (2X for 30 min each) followed by 5 ml of 1X restriction digestion buffer (2X for 1 h each) in a 6-well dish. The agarose plug was then transferred to a humidity chamber and incubated with 100 μl of restriction enzyme reaction mix at 37°C overnight. An additional 30 μl of restriction enzyme reaction mix was added to each plug and incubated for 2–3 h. The plug was then washed with TE pH 8.0 and TE containing 0.1 mM EDTA, pH 8.0 for 1 h each before 2-D gel analysis as described [18]. DNA fragments containing *ARS305*, *ARS813*, *ARS1006*, and *ARS207.1*, *ARS(13:269)*, and *ARS(16:560)* were amplified by PCR from genomic DNA and used for Southern blot. PCR primer sequences are listed in S3 Table in S2 File.

## Chromatin immunoprecipitation (ChIP)

ChIP experiments were performed with yeast strains harboring epitope tagged-CDC45 or -RAD53, separately. Log phase cells grown in YEPD medium were synchronized in G1 with α-factor and then released into S phase in the presence of 200 mM HU. 285 ml and 200 ml of cells were collected at G1 and every 20 min up to 60 min post release. ChIP DNA isolation method was adapted from the Zegerman lab protocol [59]. The collected cells were first centrifuged at 1000 x g at 4°C for 5 min, followed by resuspension in 35 ml of YEPD medium to

facilitate subsequent crosslinking. Cells containing CDC45-3X-HA were crosslinked by adding formaldehyde at 1% and incubating for 25 min, followed by quenching with 125 mM glycine and incubating for 5 min. Cells containing RAD53-2X-HA-6X-HIS were first crosslinked with 1 mM EGS (Ethylene glycol bis(succinimidyl succinate) and incubation for 10 min, then with 1% formaldehyde for additional incubation for 10 min, followed by quenching with 125 mM glycine and incubation for 25 min. The crosslinking and quenching steps were performed at room temperature with gentle mixing on a rocker. Crosslinked cells were washed with 10 ml cold 1X PBS, followed by 10 ml 50 mM HEPES. Cells resuspended in 1 ml 50 mM HEPES were then transferred to eppendorf tubes followed by centrifugation at 13,200 x rpm for 2 min at 4˚C. Cell pellets were quick-frozen on dry ice and stored at -80˚C overnight. Frozen cell pellets were thawed on ice in 300 µl lysis buffer (50 mM HEPES/KOH pH7.5, 1mM EDTA, 1% Triton X-100, 0.1% Sodium deoxycholate, 140 mM NaCl, Halt Protease and Phosphatase Inhibitor cocktail (Thermo Scientific), Halt Phosphatase Inhibitor (Thermo Scientific)). Thawed cell suspension was then transferred to 2-ml screw-cap tubes containing 300 µl glass beads. Cells were lysed in a bead beater homogenizer in the cold room, using six 30-sec cycles with 3 min rest on ice between every two cycles. Cell lysis was confirmed by phase contrast microscopy. Cell lysate was collected through the bottom of the tube by hot needle piercing and centrifugation at 1,400 x g for 3 min. Cell lysates were then sonicated using Covaris M220 Focused-Ultrasonicator for 8–10 min at 10% duty factor, 200 cycles/burst, and 75W peak power, to obtain target DNA size of 200–600 bp. Sonicated DNA cell lysate was centrifuged at 13,000 rpm for 20 min at 4˚C to remove the insoluble material. The supernatant was transferred to a fresh tube and the volume was brought up to 1 ml using lysis buffer. A 50-µl aliquot was used for agarose gel electrophoresis to check DNA size and a 40-µl aliquot was collected as "Input" control. The remaining 910 µl lysate was used for IP reaction with 80 µl anti-HA conjugated magnetic beads (Pierce, prewashed once with PBS and lysis buffer) at 4˚C on a rotator overnight. The beads were washed sequentially with 1 ml of each of the lysis buffer, wash buffer 1 (50 mM HEPES/KOH pH7.5, 1 mM EDTA, 1% Triton X-100, 0.1% Sodium deoxycholate and 250 mM NaCl), wash buffer 2 (50 mM HEPES/KOH pH7.5, 1 mM EDTA, 1% Triton X-100, 0.1% Sodium deoxycholate and 500 mM NaCl), wash buffer 3 (0.25 M LiCl, 0.5% NP-40, 0.5% Sodium deoxycholate, 1mM EDTA, 10 mM Tris-HCl pH 8), then finally, TE pH 8. DNA was eluted from the washed beads by 160 µl elution buffer (0.85X TE pH 8, 1% SDS, 0.25 M NaCl) at 65˚C for 30 min with intermittent vortexing. Eluted DNA and well as Input DNA were incubated with 8 µl DNase-free RNase (100 µg/ml) at 37˚C for 2 h, followed by incubation with 14 µl Proteinase K (10 mg/ml) at 65˚C overnight to reverse crosslinks. The resultant DNA was purified using phenol/chloroform extraction and ethanol precipitation overnight with 3 µl or 1 µl glycogen as a carrier for ChIP or Input DNA, respectively. ChIP and Input DNA were resuspended in 50 µl and 100 µl of TE 0.1 respectively.

## ChIP-qPCR

For both CDC45- and RAD53-ChIP experiments, 1 µl each of ChIP DNA and 0.1 µl (1 µl of a 1:10 dilution) of Input DNA was used as template for qPCR analysis. These optimal quantities of template DNA was determined based on standard curve measurements with serial diluted template DNA. The primer sequences are shown in S3 Table in S2 File. Template DNA was reconstituted in a 30 µl reaction with 2x iTag Universal SYBR Green Supermix (Bio-Rad) and 250 nM of each primer. Each 30 µl reaction was then split and analyzed as triplicates. The amount of Cdc45 and Rad53 in the ChIP sample in comparison to the Input was represented as "% Input", wherein % of Input = $100*2^{(Cq\ Input(corrected)–Cq\ ChIP)}$, wherein $Cq_{Input(corrected)} = (Ct_{Input}—\log_2(\text{dilution factor}))$. The % Input signals at each locus were then normalized to the

% Input signals at the control locus (*ARS306-dist*), expressed as "Relative Enrichment", for each cell sample.

## 10 RNA isolation and Real-time quantitative PCR (RT-qPCR)

Cells were grown and subjected to synchronization followed by release into S phase in the presence of 200 mM HU as described above. At each time point, 10 ml of cell culture was collected and washed with cold $H_2O$, followed by quick-freezing on dry ice and stored at -80˚C until further use. RNA isolation was performed by using the hot phenol/chloroform method [61]. The isolated total RNA was then enriched for mRNA with the RNeasy mini kit (Qiagen), which also implemented RNase-free DNase treatment (Qiagen) to remove contaminating genomic DNA. One μg of mRNA was reverse transcribed to cDNA using iScript cDNA Synthesis Kit (Bio-Rad) in a 20-μl reaction. The synthesized cDNA was then diluted 1:5 with DEPC $H_2O$ and 1 μl of the resulting mix was analyzed using iTag Universal SYBR Green Supermix (Bio-Rad) in a 10-μl reaction in triplicates. The genomic loci under examination are shown in Fig 5D and primers used are listed in S3 Table in S2 File.

## Statistical analysis

For both ChIP-qPCR as well as gene expression RT-qPCR, One-way ANOVA test followed by Tukey's multiple testing for pairwise comparison was performed to find statistical significance between the different timepoints and starins respectively. Error bars indicate standard deviation whereas the annotations for p values are: *, $p < 0.05$; **, $p < 0.01$; ***, $p < 0.001$; ****, $p < 0.0001$.

## Whole genome sequencing and *de novo* assembly of the A364a genome

A364a *RAD53* cells (KK14-3a) were grown to mid-log phase in synthetic complete medium. Cells were harvested with the addition of 0.1% sodium azide and 50 mM EDTA. Cell were prepared for FACS by treating with 0.25 mg/ml RNase A for 1 h, 1 mg/ml proteinase K for 1 h, and stained with 1 μM SYTOX Green. G2 and S-phase cells were collected using a BD FACS Aria IIu machine. Genomic DNA was prepared using the Zymo Research YeaSTAR genomic DNA kit. Whole genome DNA libraries were prepared using the Illumina Nextera DNA Kit and sequenced at 2x50 bp paired-end on Illumina HiSeq 2500. Approximately 13 million reads for the S-phase sample were obtained and used for further analysis. *De novo* sequence assembly of the A364a genome was performed with Velvet [62] using default parameters.

## Sequence analysis of strain-specific origins

Sequences for strain-specific origins, *i.e.*, origins that are activated in only one of the two genetic backgrounds (13 in A364a and 57 in W303), were obtained from the S288C reference genome R64.2.1 at the Saccharomyces Genome Database (https://downloads.yeastgenome.org/sequence/S288C_reference/genome_releases/). The range of sequences for each origin was determined based on OriDB classification. These origin sequences were then compared against the W303 contigs [15] and the A364a contigs (*de novo* assembly in this study) using the BLAST2.6.0+ engine at NCBI [63]. The top hit with sequence length difference less than 20 bp for each origin was used for further analysis. Origins with SNPs and InDels were thus identified through sequence alignment. ACSs in the 71 strain-specific origins were identified by FIMO scans from the MEME Suite [64].

## Supporting information

**S1 File. Complete list of all activated origins with their firing status in the RAD53 vs. rad53 cells in the A364a and W303 background.** Each origin is also marked by the origin category defined in the text.
(XLSX)

**S2 File. Supporting information tables and figures.**
(PDF)

**S1 Raw images.**
(PDF)

## Acknowledgments

We wish to thank M. K. Raghuraman and Bonita Brewer for helpful discussions and sharing the R script for heat maps. We are grateful to Steve Bell for p405-CDC45-HA/C, Eric Foss for pEF395, Bruce Knutson for pRS303, Patricia Kane for α-Vma1 antibody, and Brian Haarer for technical advice. We also thank Philip Zegerman for generously sharing western and ChIP protocols.

## Author Contributions

**Conceptualization:** Jie Peng, Wenyi Feng.

**Data curation:** Ishita Joshi, Jie Peng, Gina Alvino, Elizabeth Kwan.

**Formal analysis:** Jie Peng, Wenyi Feng.

**Funding acquisition:** Wenyi Feng.

**Investigation:** Jie Peng.

**Methodology:** Jie Peng.

**Project administration:** Wenyi Feng.

**Supervision:** Wenyi Feng.

**Visualization:** Ishita Joshi.

**Writing – original draft:** Jie Peng, Wenyi Feng.

**Writing – review & editing:** Ishita Joshi, Gina Alvino, Wenyi Feng.

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
