## [Decision Letter · Decision Letter 0]

13 Apr 2021

PONE-D-21-08460

Exceptional origin activation revealed by comparative analysis in two laboratory yeast strains

PLOS ONE

Dear Dr. Feng,

Thank you for submitting your manuscript to PLOS ONE. After careful consideration, we feel that it has merit but does not fully meet PLOS ONE’s publication criteria as it currently stands. Therefore, we invite you to submit a revised version of the manuscript that addresses the points raised during the review process.

All reviewers considered this study to be a significant addition to the field. However, they also point out the need for further experimental corroboration of hypotheses, further consideration of the validity of hypotheses, and additional elaboration of the findings reported in the paper.The major issues that must be remedied are noted below:The experiments need to be extended to answer some additional basic issues including the association of Rad53 to the origins. This issue should be addressed using ChIP-qPCR of initiation factors at several well-defined origins on chromosome III in the RAD53 and rad53 strains of both W303 and A364a. The genomic studies that suggest the presence of Rad53-dependent origins should be by performing RT-qPCR on RNA extracted from HU- treated RAD53 and rad53 cellsThe investigators should describe the experimental reasons for the choice of the two strains used and compare the known characteristics of these strains. The authors performed replication origin identification at the genomic level. This information must be provided to the reader in supplementary information (either directly or through a link) The authors should also use to DeOri database.The authors need to point out that autophosphorylation is inferred rather than directly shown or provide alternative data demonstrating autophosphorylation. The Remus studies refer to an in vitro analysis. Is it appropriate to apply these data to an in vivo situation? Furthermore, the authors need to modify their hypothesis for a negative regulation of fork movement given the possibility of alternative interpretations.Provide a rationale for the use of the Vma1 protein for Western blot normalization.The authors need generally to discuss their results and speculations to a greater degree in the text as discussed by the Reviewers.In Discussion, the authors need to differentiate between the dNTP effect on fork stalling vs. fork collapse.Studies were not conducted in the absence of HU to my understanding. Please eliminate indications to the contrary or provide additional data.Authors should compare the analysis from the Bielinsky lab who compared the genome-wide replication profiles of rad53-1 mutant are described in as much as the differing strain backgrounds permit. Please state the reference sequence in Figure 4.Please edit the text carefully to eliminate grammatical errors.Please address all other issues raised by the Reviewers in a point-by-point response.The Reviewers did not disagree on central issues but rather provided complementary analyses of the studies presented here.The AE has independently analyzed the paper and concur with the critiques provided by the external Reviewers. I would also like to note that the the authors provide all line numbers to any changes made in the text. This makes the process of evaluating revisions far easer.

We look forward to receiving your revised manuscript.

Kind regards,

Arthur J. Lustig, PhD

Academic Editor

PLOS ONE

Journal Requirements:

PLOS ONE now requires that authors provide the original uncropped and unadjusted images underlying all blot or gel results reported in a submission’s figures or Supporting Information files. This policy and the journal’s other requirements for blot/gel reporting and figure preparation are described in detail at https://journals.plos.org/plosone/s/figures#loc-blot-and-gel-reporting-requirements and https://journals.plos.org/plosone/s/figures#loc-preparing-figures-from-image-files. When you submit your revised manuscript, please ensure that your figures adhere fully to these guidelines and provide the original underlying images for all blot or gel data reported in your submission. See the following link for instructions on providing the original image data: https://journals.plos.org/plosone/s/figures#loc-original-images-for-blots-and-gels.

Reviewers' comments:

Reviewer's Responses to Questions

**Comments to the Author**

1. Is the manuscript technically sound, and do the data support the conclusions?

Reviewer #1: Yes

Reviewer #2: Yes

Reviewer #3: Yes

2. Has the statistical analysis been performed appropriately and rigorously? 

Reviewer #1: I Don't Know

Reviewer #2: Yes

Reviewer #3: N/A

3. Have the authors made all data underlying the findings in their manuscript fully available?

Reviewer #1: Yes

Reviewer #2: No

Reviewer #3: Yes

4. Is the manuscript presented in an intelligible fashion and written in standard English?

Reviewer #1: Yes

Reviewer #2: Yes

Reviewer #3: Yes

5. Review Comments to the Author

Reviewer #1: In this manuscript, the authors used a published single-stranded DNA mapping protocol to identify active replication origins in two different laboratory strains of S. cerevisiae (W303 and A364a) after their release from G1 into hydroxyurea in the presence or absence of an active Rad53 S-phase checkpoint factor. The majority of active origins were shared between the two strains and these origins also shared the same overall response to the status of the Rad53-dependent S phase checkpoint. However, the level of ssDNA at origins and the distance traveled by the fork were significantly more reduced in the W303 rad53 mutant strain and correlated with a more robust phosphorylation of Rad53 in the W303 RAD53 strain. A number (71) of origins were only active in W303 or A364a. Comparison of published genome DNA sequencing data between the two strains found that sequence polymorphisms in the consensus ACS element were unlikely to account for the differences. The authors believed that origin activation likely occurred because of the presence of multiple mismatched ACS elements in the DNA sequences. Finally, the authors identified 6 origins that they characterized as Rad53-dependent, in that they were only active in RAD53 but not in rad53 strains. Interestingly, 5 of these origins also overlapped with an ORF, in contrast to the intergenic location of most origins. The lack of origin activation in rad53 cells was not a consequence of transcriptional interference, as published data indicated that the genes in which the origins were located were transcribed at a lower level in rad53 cells compared to RAD53 cells.

Overall, this is an interesting and well documented study, with many of the conclusions from genomic data validated by 2D agarose gel analysis. One of the most remarkable findings was the observation that despite the similar number of origins activated in rad53 between the two strains, W303 was significantly more sensitive to the reduced checkpoint than A364a, showing reduced levels of ssDNA at all origins and reduced spreading of ssDNA from origins. This is shown most dramatically in Figure 1C. The authors speculated that this might be due to the lower level of Rad53 in the W303 rad53 mutant, which in turn led to lower levels of Rad53-P. The striking differences between the two strains need to be much more fully discussed in the text. What could account for the lower activation of all origins in the W303 rad53 mutant? Is this a consequence of the role of Rad53 in the initiation of DNA replication that is independent of checkpoint regulation, and if so, by what mechanism? Is the loading of initiation factors altered at origins, and is Rad53 binding to origins affected? Is there an effect on nucleotide pools in this particular rad53 mutant? Some of these questions could be addressed using ChIP-qPCR of initiation factors at several of the well defined origins on chromosome III in the RAD53 and rad53 strains of both W303 and A364a. This would help to address questions on the differential effects of the rad53 mutation in the two strains, and perhaps more generally uncover some new insights into the relationship of the levels of Rad53 to replication initiation.

Other comments:

1. The identification of RAD53-dependent origins in 5 ORFs represents a very interesting finding. The authors should confirm the published genomic data on these genes by performing RT-qPCR on RNA extracted from HU treated RAD53 and rad53 cells. Some speculation on the possible function of RAD53 at these genes would also be beneficial.

2. The text needs to be carefully edited as there were some missing sentence parts.

Reviewer #2: Identification and characterization of replication origins are essential for a better understanding of the molecular mechanism of DNA replication. The authors conducted experimental procedures to perform the differential dynamics of origin activation in the A364a and W303 Saccharomyces cerevisiae strains. The authors also found the groups of “Rad53-unchecked” and “Rad53-checked” origins by the cooperation of origin usage in wild type and the rad53 mutant cells. The identification of replication origins from the genome-wide and the analysis of a new class of origins would provide new insights into the replication mechanism of S. cerevisiae.

Major comments:

1.Authors should describe the detailed reasons for choosing the strains used in the study. What’s the phylogenetic distance of A364a and W303 S. cerevisiae strains? The authors mentioned that “Genetic variation in diverse laboratory strains can manifest in distinct physiological properties”. What’s the phenotypic difference between these two strains? What’s the significant phenotypic difference associated with DNA replication between these two strains?

2.The authors identified the “Rad53-dependent origins”, however, the author needs to clarify that these “Rad53-dependent origins” are not strain-specific.

Minor comments:

1.The authors used the ARS records of the OriDB database. OriDB is one of the well-known yeast replication origin databases. I recommended the authors also use the DeOri (DOI: 10.1093/bioinformatics/bts151), a database for eukaryotic replication origins, to verify and support your experimental results.

2.The authors performed the identification of replication origins of two yeast strains by genome-wide level. I’d like to suggest the authors attach the detailed information of identified replication origins, including chromosomal position and replication origin sequences, to the supplementary file, which will help the authors and other researchers to further explore the mechanism of DNA replication.

3.I’d like to recommend some of recent papers related to DNA replication origins in Saccharomyces cerevisiae genome for your kind reference.

Wang, D, Lai FL, Gao, F. Ori-Finder 3: a web server for genome-wide prediction of replication origins in Saccharomycescerevisiae. Briefings in Bioinformatics 2020, doi: 10.1093/bib/bbaa182

Wang D, Gao F. Comprehensive Analysis of Replication Origins in Saccharomyces cerevisiae Genomes. Frontiers in Microbiology, 2019, 10: 2122.

Peng C, Luo H, Zhang X, et al. Recent advances in the genome-wide study of DNA replication origins in yeast. Frontiers in Microbiology, 2015, 6(FEB): 117.

Reviewer #3: Comments on the manuscript PONE-D-21-08460 (Peng et al. entitled ‘Exceptional origin activation revealed by comparative analysis in two laboratory yeast strains’).

In this manuscript, authors performed a comparative analysis of replication origin activation in two yeast strains W303 and A364a when cells were exposed with hydroxyurea. They also analyzed the effect on origin activation of the checkpoint kinase, Rad53, by combining the checkpoint-deficient Rad53-K227A mutant. They found that the activation of replication origins are similar in both strains. However, they find strain-specific origin usage. Although some of strain-specific origins have SNPs, they suggested that the SNP is not the reason of the strain-specificity. They also suggested that the difference of origin usage is partly depends on the activity of Rad53. Finally, they identified a new class of origins that are active only when Rad53 is functional.

Although the study is descriptive rather than analytical, some of their findings are interesting and are worth for the publication in the PLoS ONE. However, I feel there are some points that should be clarified before acceptance.

Major points

1. Authors insist that the Rad53 levels are different between W303 and A364a (p. 10). It does not seem that there is a big difference between wildtypes. I also wonder why authors use Vma1 as a control rather than ponceau staining. Moreover, it is known that Rad53 autophosphorylates when it is activated. If authors want to indicate the difference of Rad53 activity in this situation, kinase assay would be the best way. The way simply compare band intensities might be inappropriate, because some of them are not autophosphorylated, it is difficult to extract only autophosphorylated ones, and the extent of autophosphorylation must be considered if you want to say the activity.

In discussion, they suggested that Rad53 has the function on fork progression by referring the recent Remus Lab’s results (ref #35). However, this is an in vitro results obtained by adding the aphidicolin to inhibit DNA polymerase function. I wonder this situation can be applied directly to cells treated with HU. Moreover, in W303 rad53-K227A cells, which has lower levels of Rad53 protein than corresponding A364a cells, fork did not travel longer than wild type nor A364a rad53-K227A (Fig. 2D). Therefore, their interpretation (p10 middle part) and discussion (p17 middle part) seem incorrect for me.

2. Authors say, ‘This is consistent with the notion that checkpoint failure causes replication forks to collapse shortly after initiation from the origin.’ (p9, bottom). I think replication forks stall rather than collapse by dNTP shortage in HU-treated cells even in checkpoint-deficient cells. If there are evidences that support authors description, they should be clearly shown here. Shorter fork travel could be caused by more origin firing. Simply because less dNTPs are available when more origins fire.

3. Authors say, ‘We conducted a comparative analysis of origin activation, with and without DNA replications tress by HU, using genomic ssDNA mapping in two common laboratory yeast strains.’ (the first sentence of Discussion). I do not think this is the case. Authors did not do anything without HU.

4. Authors do not refer the previous analysis form Bielinsky lab (Raveendranathan et al. (2006) EMBO J. 25:3627-39), in which genome-wide replication profiles in rad53-1 mutant are described. If authors can compare both results, it would be great. I admit this might be difficult, because the strain background is different (Bielinsky lab uses the strain BF-15D).

5. Authors concluded that SNPs cannot explain the strain-specific origin activation. Authors can obtain solid conclusion by introducing such SNPs into other strains directly to see the effect of SNPs.

Minor points

In Figure 4, authors show the difference in the DNA sequence in the strains. What is the reference sequence? S288c in the Saccharomyces Genome Database, or not?

Some sentences are broken. For example,

Page 5, lines 18-19.

Page 16, line 3. ‘replications tress’ should be ‘replication stress’

There could be more.

6. PLOS authors have the option to publish the peer review history of their article (what does this mean?). If published, this will include your full peer review and any attached files.

Reviewer #1: No

Reviewer #2: No

Reviewer #3: No

---

## [Author Response · Author response to Decision Letter 0]

13 Oct 2021

Response to reviews have been uploaded as a separate document.

---

## [Decision Letter · Decision Letter 1]

9 Nov 2021

PONE-D-21-08460R1Exceptional origin activation revealed by comparative analysis in two laboratory yeast strainsPLOS ONE

Dear Dr. Feng,

Thank you for submitting your manuscript to PLOS ONE. After careful consideration, we feel that it has merit but does not fully meet PLOS ONE’s publication criteria as it currently stands. Therefore, we invite you to submit a revised version of the manuscript that addresses the points raised during the review process.

The revised manuscripts requires further changes before publication is possible. This conclusion is drawn from the current  limitations in the interpretation of the new data, the absence of some statistical data, and some lack of clarity in defining some terms. These issues are addressed by Reviewer 3 and the AE fully concurs..The first and most significant issue concerns the methods used for the newly included ChIP data which apparently used PCR, rather than qPCR, data. qPCR MUST be used for ChIP data by standard conventions.  This may also explain why the data is inconsistent with published accounts.These experiments must be redone before revision.The second issue is the some weakness in the statistical analysis as described by Reviewer 3.The third issue is a lack of distinction between replication form collapse and stalling. Please define these terms more carefully.Finally, answer all of the other sundry questions raised by Reviewer's 2 and 3. Please ensure that your decision is justified on PLOS ONE’s publication criteria and not, for example, on novelty or perceived impact.

We look forward to receiving your revised manuscript.

Kind regards,

Arthur J. Lustig, PhD

Academic Editor

PLOS ONE

Journal Requirements:

Reviewers' comments:

Reviewer's Responses to Questions

**Comments to the Author**

1. If the authors have adequately addressed your comments raised in a previous round of review and you feel that this manuscript is now acceptable for publication, you may indicate that here to bypass the “Comments to the Author” section, enter your conflict of interest statement in the “Confidential to Editor” section, and submit your "Accept" recommendation.

Reviewer #1: All comments have been addressed

Reviewer #2: All comments have been addressed

Reviewer #3: (No Response)

2. Is the manuscript technically sound, and do the data support the conclusions?

Reviewer #1: Yes

Reviewer #2: Yes

Reviewer #3: Partly

3. Has the statistical analysis been performed appropriately and rigorously? 

Reviewer #1: Yes

Reviewer #2: Yes

Reviewer #3: No

4. Have the authors made all data underlying the findings in their manuscript fully available?

Reviewer #1: Yes

Reviewer #2: Yes

Reviewer #3: Yes

5. Is the manuscript presented in an intelligible fashion and written in standard English?

Reviewer #1: Yes

Reviewer #2: Yes

Reviewer #3: Yes

6. Review Comments to the Author

Reviewer #1: (No Response)

Reviewer #2: Almost all the comments have been addressed properly. However, I still have some minor comments as follows.

1. DeOri database was mentioned in the main text of the manuscript, but the relevant reference was not cited.

2. Some words in Figure 6D were obscured, such as ARS(16:560), 4.5 on the Y axis.

3. The words A364a and W303 shall be of uniform size in Figure 1B.

Reviewer #3: Comments on the manuscript PONE-D-21-08460R1 (Joshi et al. entitled ‘Exceptional origin activation revealed by comparative analysis in two laboratory yeast strains’).

In this revised version, authors addressed most of issues raise by me. I also appreciate authors’ effort to improve their work. However, I feel some issues are not fully addressed in a satisfactory manner. Moreover, I noticed that newly added data include other issues that cannot be published as it is. Specific points are follows.

1. (Previous Major point #1) Authors responded as ‘We agree with the reviewer that the difference between Rad53 levels in WT strains is moderate. Yet they were reproducibly seen.’ If it is reproducible, authors should show the average of Rad53/Vma1 value obtained from several repeats and statistical (p) value. Authors also need to explain why Vma1 is appropriate as an internal control.

2. (Previous Major point #2) Authors seems not to distinguish ‘stall’ and ‘collapse’ rigorously. In my understanding, when dNTPs short, first, replication forks stall. If cells do not have proper checkpoint machineries at this point, integrity of replisome then will be lost and replication fork cannot be restart anymore (= collapse) even after the supply of dNTP is back, as shown in Labib lab. In addition to this, If CMG itself is collapse by accident, replication fork will be collapsed. In this manuscript, most of ss region detected by authors in rad53 mutant are first caused by fork stall, rather than collapse of the replication fork, I guess.

3. Newly added ChIP data in Figure 3 do not agree with previous data obtained in many labs (Araki, Bell, Labib labs etc.). That is: in G1 phase, Cdc45 does associate with early origins but neither late origins nor non-origin region. In this experiment, authors do not include negative control and use conventional PCR rather than qPCR to measure the amount of ChIP-ed DNA. This might be the cause of the discrepancy. The way to show results in this figure are also not acceptable, because all statistical descriptions including that on error bars are missing. I also am not sure Rad53 can ChIP replication origins. No one have shown such association, as far as I know.

4. Figure 6D. As in Figure 3 all statistical details are missing, and I cannot understand why authors put a break in Y axis. What does it mean the value of Y axis: ‘relative expression (S/G1)’? I cannot find any explanation in the manuscript. Error bar in P1 (red) from ARS(16:560) which reaches to 5, seems to be a bad joke.

5. (just a comment) ‘Revised Manuscript with Track Changes’ does not include proper ‘track-change’ info, which made review process more laborious.

7. PLOS authors have the option to publish the peer review history of their article (what does this mean?). If published, this will include your full peer review and any attached files.

Reviewer #1: No

Reviewer #2: No

Reviewer #3: No

---

## [Author Response · Author response to Decision Letter 1]

24 Dec 2021

Please see attached document containing our response to reviews.

---

## [Decision Letter · Decision Letter 2]

19 Jan 2022

PONE-D-21-08460R2Exceptional origin activation revealed by comparative analysis in two laboratory yeast strainsPLOS ONE

Dear Dr. Feng,

Thank you for submitting your manuscript to PLOS ONE. After careful consideration, we feel that it has merit but does not fully meet PLOS ONE’s publication criteria as it currently stands. Therefore, we invite you to submit a revised version of the manuscript that addresses the points raised during the review process.All of the required changes have been satisforilly made with the exception of the Figure error pointed out by the Reviewer.  Please fix this issue and return the manuscript for final approval.Please submit your revised manuscript by Mar 05 2022 11:59PM. If you will need more time than this to complete your revisions, please reply to this message or contact the journal office at plosone@plos.org. Please include the following items when submitting your revised manuscript:A rebuttal letter that responds to each point raised by the academic editor and reviewer(s). You should upload this letter as a separate file labeled 'Response to Reviewers'.A marked-up copy of your manuscript that highlights changes made to the original version. You should upload this as a separate file labeled 'Revised Manuscript with Track Changes'.An unmarked version of your revised paper without tracked changes. You should upload this as a separate file labeled 'Manuscript'.If applicable, we recommend that you deposit your laboratory protocols in protocols.io to enhance the reproducibility of your results. Protocols.io assigns your protocol its own identifier (DOI) so that it can be cited independently in the future. For instructions see: https://journals.plos.org/plosone/s/submission-guidelines#loc-laboratory-protocols. Additionally, PLOS ONE offers an option for publishing peer-reviewed Lab Protocol articles, which describe protocols hosted on protocols.io. Read more information on sharing protocols at https://plos.org/protocols?utm_medium=editorial-email&utm_source=authorletters&utm_campaign=protocols.

We look forward to receiving your revised manuscript.

Kind regards,

Arthur J. Lustig, PhD

Academic Editor

PLOS ONE

Journal Requirements:

Reviewers' comments:

Reviewer's Responses to Questions

**Comments to the Author**

1. If the authors have adequately addressed your comments raised in a previous round of review and you feel that this manuscript is now acceptable for publication, you may indicate that here to bypass the “Comments to the Author” section, enter your conflict of interest statement in the “Confidential to Editor” section, and submit your "Accept" recommendation.

Reviewer #3: (No Response)

2. Is the manuscript technically sound, and do the data support the conclusions?

Reviewer #3: Yes

3. Has the statistical analysis been performed appropriately and rigorously? 

Reviewer #3: Yes

4. Have the authors made all data underlying the findings in their manuscript fully available?

Reviewer #3: Yes

5. Is the manuscript presented in an intelligible fashion and written in standard English?

Reviewer #3: Yes

6. Review Comments to the Author

Reviewer #3: Comments on the manuscript PONE-D-21-08460R2 (Joshi et al. entitled ‘Exceptional origin activation revealed by comparative analysis in two laboratory yeast strains’).

In the left panel of Figure 6D (a bar graph of ChIP results on ARS(13:269), authors break graph just above 4 and show only one scale in the upper part of the graph. As a result, it is not possible to read the values of several results and the top of the bar of P1 from A364a_rad53 are not shown. These issues must be fixed before publication.

7. PLOS authors have the option to publish the peer review history of their article (what does this mean?). If published, this will include your full peer review and any attached files.

Reviewer #3: No

---

## [Author Response · Author response to Decision Letter 2]

20 Jan 2022

Response to reviews have been uploaded as a separate document.

---

## [Editor Report · Decision Letter 3]

24 Jan 2022

Exceptional origin activation revealed by comparative analysis in two laboratory yeast strains

PONE-D-21-08460R3

Dear Dr. Feng,

We’re pleased to inform you that your manuscript has been judged scientifically suitable for publication and will be formally accepted for publication once it meets all outstanding technical requirements.

Kind regards,

Arthur J. Lustig, PhD

Academic Editor

PLOS ONE
---

## [Editor Report · Acceptance letter]

4 Feb 2022

PONE-D-21-08460R3 

Exceptional origin activation revealed by comparative analysis in two laboratory yeast strains 

Dear Dr. Feng:

I'm pleased to inform you that your manuscript has been deemed suitable for publication in PLOS ONE. Congratulations! Your manuscript is now with our production department. 

Kind regards, 

on behalf of

Dr. Arthur J. Lustig 

Academic Editor

PLOS ONE